# Endogenous Neural Stem Cell Mediated Oligodendrogenesis in the Adult Mammalian Brain

**DOI:** 10.3390/cells11132101

**Published:** 2022-07-02

**Authors:** Daniel Z. Radecki, Jayshree Samanta

**Affiliations:** Department of Comparative Biosciences, School of Veterinary Medicine, Stem Cell and Regenerative Medicine Center, University of Wisconsin-Madison, Madison, WI 53706, USA; dradecki@wisc.edu

**Keywords:** oligodendrocyte, oligodendrogenesis, adult neural stem cell, subventricular zone, subgranular zone, tanycyte, cerebellum, Bergmann glia

## Abstract

Oligodendrogenesis is essential for replacing worn-out oligodendrocytes, promoting myelin plasticity, and for myelin repair following a demyelinating injury in the adult mammalian brain. Neural stem cells are an important source of oligodendrocytes in the adult brain; however, there are considerable differences in oligodendrogenesis from neural stem cells residing in different areas of the adult brain. Amongst the distinct niches containing neural stem cells, the subventricular zone lining the lateral ventricles and the subgranular zone in the dentate gyrus of the hippocampus are considered the principle areas of adult neurogenesis. In addition to these areas, radial glia-like cells, which are the precursors of neural stem cells, are found in the lining of the third ventricle, where they are called tanycytes, and in the cerebellum, where they are called Bergmann glia. In this review, we will describe the contribution and regulation of each of these niches in adult oligodendrogenesis.

## 1. Introduction

In the central nervous system, the main function of oligodendrocytes is to myelinate axons by spirally wrapping their cellular processes around them. Myelination is essential to increase the speed of conduction of action potentials along the axons and results in devastating functional consequences when it is lost in demyelinating diseases such as multiple sclerosis (MS). Oligodendrogenesis, which is the generation of new oligodendrocytes, is essential in the healthy adult brain to replace worn-out cells and for adaptive myelination in the healthy brain. It is also required for regenerating myelin or remyelination to repair the axons that have lost their myelin or are demyelinated as a result of various neurological diseases. In the healthy adult mouse brain, 90% of oligodendrocytes survive for more than 8 months, indicating a very low rate of oligodendrogenesis [1]. Similarly, the oligodendrocyte population in the white matter of the human brain is remarkably stable, with an annual turnover of 0.3%, after the completion of developmental myelination [2]. Hence, a major role of oligodendrogenesis is regeneration of myelin in the diseased brain. The most common neurological disease characterized by loss of myelin is multiple sclerosis (MS), but recent studies have highlighted the presence of demyelination in other neurodegenerative diseases such as Alzheimer’s disease, Amyotrophic lateral sclerosis, Huntington’s disease, and Parkinson’s disease [3]. Thus, regeneration of myelin is likely to be a common therapeutic strategy for ameliorating or slowing the disease course of several neurological diseases.

Myelin can be regenerated either by adult oligodendrocytes in the surrounding area [4,5] or by newly formed oligodendrocytes produced in response to a demyelinating injury. However, recent studies in zebrafish suggest that in contrast to new oligodendrocytes, the surviving old oligodendrocytes not only elaborate fewer myelin sheaths but also target the myelin to neuronal cell bodies. Such myelin-wrapped neuronal cell bodies are also observed in postmortem human MS brains, implying that remyelination by existing oligodendrocytes is highly inefficient [6].

There are two sources of new oligodendrocytes in the adult mammalian brain—oligodendrocyte progenitor cells (OPCs), which are present throughout the parenchyma, and neural stem cells (NSCs), which reside in specific niches in the brain. Although OPCs are an important source of new oligodendrocytes in demyelinated lesions [7,8,9,10,11,12,13,14], for the purpose of this review, we will focus on regeneration of oligodendrocytes from NSCs in the adult mammalian brain, after the completion of developmental myelination.

NSCs differentiate into oligodendrocytes by transitioning through various stages of development that include the OPCs, marked by expression of NG2 and PDGFRα, which ultimately differentiate into mature oligodendrocytes expressing CC1 and myelin proteins such as MBP and MOG both in vivo and in vitro [8]. In the adult brain, they can promote remyelination in vivo either by replenishing the adult OPCs recruited into lesions, or as a direct source of oligodendrocytes themselves. NSCs are defined as multipotent cells that (i) express stem cell markers such as GFAP, GLAST, and CD133; (ii) are capable of self-renewing; and (iii) have the capacity to differentiate into neurons, astrocytes, and oligodendrocytes at least in vitro. Amongst the stem cell niches in the adult mammalian brain, the subventricular zone (SVZ) lining the walls of the lateral ventricles and the subgranular zone (SGZ) in the dentate gyrus of the hippocampus are the primary zones for neurogenesis. However, other areas such as the cerebellum and the lining of the third ventricle, that harbor radial glia-like cells, may have the potential for generation of oligodendrocytes in vivo [15]. In this review, we will describe the role of NSCs in oligodendrogenesis, in each of these areas.

## 2. Subventricular Zone (SVZ)

The SVZ, also called the ventricular–subventricular zone (V-SVZ) and subependymal zone (SEZ), is the largest germinative area and includes the medial, lateral, and dorsal or subcallosal walls of the lateral ventricles. The adult SVZ consists of slowly dividing, quiescent type B NSCs (qNSCs) expressing GFAP, GLAST, and CD133, which upregulate nestin and EGFR when activated for neurogenesis. These activated NSCs divide to produce the type C transit-amplifying progenitors (TAPs), expressing nestin, Ascl1 (Mash1), Olig2, Dlx2, and EGFR [16,17,18,19,20,21]. The TAPs in turn generate the type A migratory neuroblasts, expressing PSA-NCAM, Pax6, and doublecortin (Dcx), which enter the rostral migratory stream (RMS) and migrate to the olfactory bulb, where they differentiate into interneurons. The qNSCs account for 3–9% of the cells in the adult SVZ and are specified from radial glial cells in the embryonic brain between E13.5 and E15.5 but remain quiescent until activated for neurogenesis in the postnatal brain [22,23,24,25,26]. They also retain their radial glia-like morphology with a primary apical cilium contacting the CSF and a longer basal process towards the blood vessels [27]. They are the least proliferative cells in the SVZ, with a cell cycle length of 17–18 h and a short S-phase length of 4 h. In contrast, the TAPs constitute 14–22% of cells in the adult SVZ [28] and are the most proliferative cells, with a cell cycle length of 18–25 h and a long S-phase length of 14–17 h, while the neuroblasts have a cell cycle length of 18 h and S-phase length of 9 h. Thus, the qNSCs divide initially to produce TAPs, which divide three times, producing the neuroblasts that undergo one or possibly two divisions before differentiating into mature postmitotic cells [29]. The aNSCs and TAPs grow in vitro as neurospheres in suspension cultures and can differentiate into all three lineages: neurons, oligodendrocytes, and astrocytes (Figure 1) [25,30]. However, the neuroblasts do not form neurospheres and can only differentiate into neurons, when cultured in vitro [31]. Of the 2–6% of proliferating aNSCs in the adult SVZ, only about 20–30% self-renew by dividing symmetrically, and the rest divide to generate TAPs, gradually depleting the pool over time [25,32,33,34,35]. However, the number of qNSCs is tightly regulated and dependent not only on the number of TAPs but also on the niche occupancy such that increasing the proliferation of TAPs results in decreasing the proliferation and numbers of qNSCs [36,37,38]. This is controlled in part by epidermal growth factor (EGF) and notch signaling, which promote their quiescence, and bone morphogenetic protein (BMP) signaling, which inhibits their activation [36,39]. The different stages of NSCs, marked by expression of GFAP, Nestin, EGFR, and PSA-NCAM, are also observed in the adult human SVZ lining the lateral ventricles [40,41,42]. Similar to the mouse NSCs, the cells from the adult human SVZ grow as neurospheres and differentiate into all three lineages: neurons, astrocytes, and oligodendrocytes in vitro [43].

The developmental origins of NSCs result in heterogeneous populations of qNSCs, which produce different types of olfactory bulb interneurons in the adult brain. Distinct regions of the germinal zones in the embryo give rise to the different subsets of qNSCs in the adult SVZ. The embryonic pallium forms the dorsal or subcallosal wall of the adult lateral ventricle, the lateral ganglionic eminence forms the dorsal part of the lateral wall, the medial ganglionic eminence forms the ventral part of the lateral wall, while the septum forms the medial wall of the adult lateral ventricles [44]. Thus, in addition to GFAP, GLAST, and CD133, which are expressed in all qNSCs, these cells express one of the transcription factors retained from their embryonic origins, e.g., Nkx2.1, Gli1, Gsx2, Emx1, and Zic [44,45,46]. Each of these subsets of qNSCs is likely to have a different potential for oligodendrogenesis in the healthy and injured adult brain.

### 2.1. Evidence of Oligodendrogenesis from NSCs in the Adult SVZ

#### 2.1.1. Lineage Tracing and Proliferation

Retroviral lineage tracing studies in neonatal rat brains provided the earliest evidence of oligodendrogenesis in the postnatal SVZ. Proliferating NSCs were labeled with stereotactic injection of retroviruses expressing LacZ into the lateral ventricles of postnatal Day 2 or 3 rats, and LacZ+ oligodendrocytes were detected a month later in the cortex, corpus callosum, and striatum [47]. Subsequent studies showed that NSCs isolated from the adult SVZ generate oligodendrocytes when grown as neurospheres in vitro [48,49,50].

Indirect evidence of oligodendrogenesis in the adult SVZ came from proliferation studies following demyelination in the mouse brain. These studies used tritiated thymidine (3H-thymidine) and 5-bromo-2′-deoxyuridine (BrdU) labeling to show an increase in proliferation of NSCs along with labeled oligodendrocytes co-expressing PSA-NCAM in the white matter corpus callosum [51,52]. Focal demyelination also increased the proliferation of TAPs and the expression of pro-oligodendrogenic transcription factors such as Olig2, Ascl1, and Nkx2.2 [53]. In fact, mild disruption of myelin was sufficient to increase proliferation of NSCs in the SVZ, resulting in enhanced oligodendrogenesis in Plp-null mice, which show progressive demyelination and axonal pathology [54]. Similarly, there was a 3-fold increase in proliferation in the SVZ, along with oligodendrocyte progenitors co-expressing PSA-NCAM in postmortem human MS brains [55]. Interestingly, this increase in oligodendrogenesis in the MS brain was associated with a decrease in neurogenesis, suggesting a fate switch as a likely mechanism along with an increase in proliferation [56].

Confirmation of oligodendrogenesis from qNSCs in the healthy adult mouse brain came from in vivo studies using avian retroviruses (RCAS) to label GFAP+ NSCs expressing the avian leukosis virus receptor Tva [50]. In these experiments, oligodendrocytes derived from qNSCs were found in the corpus callosum, striatum, and fimbria, and in vitro clonal analysis further confirmed the generation of oligodendrocytes from these NSCs. In addition, there was a 4-fold increase in oligodendrocytes derived from qNSCs in the corpus callosum following lysolecithin-induced focal demyelination [50]. Similarly, lineage tracing of Nestin+ NSCs and TAPs following demyelination revealed that these cells migrate to the lesion and generate myelinating oligodendrocytes predominantly in the area of the corpus callosum adjacent to the anterior SVZ. However, this response of NSCs in the SVZ is delayed and occurs after the exhaustion of oligodendrogenesis by local OPCs [57,58]. Interestingly, all qNSCs do not have the same potential for oligodendrogenesis. Although all NSCs are multipotent, live imaging of the NSCs derived from the dorsal and lateral SVZ showed that a single NSC does not give rise to both neurons and oligodendrocytes, and that expression of Pax6 or Olig2 in TAPs determines their neurogenic vs. oligodendrogenic fate, respectively [59]. Moreover, oligodendrogenesis is regionalized, with the dorsal SVZ being more oligodendrogenic compared to the more neurogenic lateral SVZ [59]. This regionalization arises from the developmental origin of oligodendrocytes.

#### 2.1.2. Developmental Origin of Adult NSC Heterogeneity

In the developing forebrain, oligodendrocytes arise in three sequential waves from discrete regions of the embryonic SVZ expressing different transcription factors, i.e., from Nkx2.1+ NSCs in the medial ganglionic eminence, followed by Gsx2+ NSCs in the lateral ganglionic eminence, and finally the Emx1+ NSCs in the pallium or cortical ventricular zone. The adult SVZ retains the Nkx2.1+ NSCs in the ventral tip, Gsx2+ NSCs in the dorsolateral wall, and the Emx1+ NSCs in the dorsal or subcallosal wall of the lateral ventricles. The Emx1- and Gsx2-derived oligodendrocytes survive in the adult brain, while the Nkx2.1-derived oligodendrocytes disappear from the postnatal brain [60,61,62]. Further, tracing the lineage of Emx1+ NSCs in the early postnatal brain showed that the Emx1-derived oligodendrocytes, preferentially remyelinated lesions following focal demyelination in the adult brain [63]. Whether the adult Emx1 pool generates new oligodendrocytes in response to demyelination remains unknown. More recently, single-cell RNAseq analysis of the adult SVZ indicated that the dorsal qNSCs have a greater potential for oligodendrocyte differentiation than the ventral qNSCs [46]. Thus, it seems likely that the dorsal SVZ has the potential to regenerate oligodendrocytes following demyelination more efficiently than the lateral SVZ.

In the adult brain, 9.2% of the cells in the dorsolateral SVZ express Gsx2; of these, 16% are GFAP+ qNSCs, and the rest are EGFR+ TAPs. Loss of Gsx2 in the adult SVZ has no effect on oligodendrogenesis, but gain of function inhibits neurogenesis and increases astrocyte generation in the olfactory bulb [64]. This is in contrast to the embryonic SVZ, where loss of Gsx2 increases differentiation into OPCs, and high levels of Gsx2 inhibit oligodendrogenesis [65]. However, further studies are needed to examine whether Gsx2+ NSCs are capable of regenerating oligodendrocytes following demyelination and whether inhibition of Gsx2 can enhance remyelination in the adult brain.

Another subset of NSCs expressing Gli1 resides in the ventrolateral wall of the lateral ventricles but does not generate oligodendrocytes in the healthy adult mouse brain; instead, they give rise to interneurons and astrocytes in the olfactory bulb (Figure 2). About 25% of Gli1-expressing cells in the SVZ are GFAP+ qNSCs, 20% are Olig2+ TAPs, and 18% are PSA-NCAM+ neuroblasts [66]. Gli1 is expressed in qNSCs and their immediate precursor cells, but not in OPCs or mature oligodendrocytes in the adult brain. Although fate-mapping the Gli1 NSCs for up to 6 months did not label oligodendrocytes [42], long-term fate-mapping detected a few oligodendrocytes in the corpus callosum, a year after labeling the Gli1 NSCs [66]. These few oligodendrocytes are likely a response to age-related loss of myelin, especially since Gli1 NSCs respond to demyelination by generating oligodendrocytes in the adult corpus callosum [42,67,68]. Loss of Gli1 further enhances oligodendrogenesis in response to demyelination [42].

Of all the different subsets of qNSCs, the Gli1 pool has been most extensively studied for its remyelination potential. Further studies are needed to confirm the role of other qNSCs such as the Emx1, Gsx2, Zic1, and Nkx2.1 subsets for oligodendrogenesis in the adult forebrain, both in health and disease.

### 2.2. Signaling Pathways Regulating Oligodendrogenesis from NSCs in the SVZ

There are several signaling pathways important for maintaining the SVZ niche and for generating oligodendrocytes in response to a demyelinating injury. Here, we have highlighted the role of the major signaling pathways: Shh, TGFβ, EGF, Wnt, and Notch pathways.

#### 2.2.1. Sonic Hedgehog (Shh)

Shh signaling is essential for maintaining the NSCs in the adult SVZ. Shh signals by binding to its receptor patched, relieving the inhibition of Smoothened (Smo), which in turn increases the activation of downstream transcription factors Gli1 and Gli2 and inhibits Gli3 repressor function [69]. In the healthy adult SVZ, ablation of Smo in all qNSCs and TAPs inhibits their proliferation and consequently reduces the generation of olfactory bulb interneurons [70]. Conversely, in vitro treatment of primary SVZ NSCs with Shh increases Gli1 expression along with their proliferation and blocks their differentiation [71,72]. Indeed, Gli1 upregulates a group of ‘stemness’ genes, including Nanog, Sox2, Klf4, Nestin, Prominin1 (CD133), and Bmi1 [73].

In the adult SVZ, Gli2 and Gli3 are expressed in all qNSCs, with slightly more abundance of Gli3 in the dorsolateral SVZ, but Gli1 is restricted to the ventrolateral SVZ. All three Gli genes are downregulated as the NSCs proliferate and progress through the TAP and neuroblast stages [74,75]. Inhibition of the Shh pathway by loss of Smo reduces the proliferation and number of qNSCs. However, this effect is rescued by the combined loss of Gli3 and Smo, indicating that Gli3 repressor function is a key inhibitor of neurogenesis in the healthy adult brain [74]. Consistently, ablation of Gli3 in the dorsolateral SVZ, which has low levels of Shh signaling, leads to an increase in Olig2+ TAPs [75]. Since Olig2 not only specifies TAPs at basal levels but also increases oligodendrogenesis when overexpressed, these studies suggest that loss of Gli3 may enhance oligodendrogenesis [28,76,77]. Further studies are needed to directly assess the effect of Gli3 on the number of mature oligodendrocytes in the corpus callosum during homeostasis and remyelination.

While loss of Smo specifically in Gli1 NSCs in the ventrolateral SVZ does not alter oligodendrogenesis following demyelination, constitutive activation of Smo in Gli1 NSCs increases proliferation of Gli1 NSCs along with the generation of OPCs without increasing the number of remyelinating oligodendrocytes [42]. In contrast, inhibition of Gli1 either genetically or pharmacologically not only enhances the proliferation of Gli1 NSCs but also increases the generation of remyelinating oligodendrocytes following demyelination [42]. Since Gli1 is expressed in response to the highest levels of Shh, these studies indicate that while Shh is important for maintenance of qNSCs and generation of OPCs, high Shh signaling inhibits the generation of mature myelinating oligodendrocytes. Gli2 is essential for the enhanced remyelination by Gli1 inhibition [78], further confirming that lower levels of Shh signaling enhance remyelination, and highlighting the nuanced role of Shh signaling in remyelination. Whether Shh plays a role in oligodendrogenesis by other subsets of qNSCs in the adult SVZ remains to be studied.

#### 2.2.2. Transforming Growth Factor-Beta (TGFβ) Superfamily

The TGFβ superfamily consists of more than 100 proteins, including the bone morphogenetic proteins (BMPs) and transforming growth factor-beta (TGFβ) cytokines. Of the more than 20 BMP family members, BMP2 and BMP4 are the most extensively studied with respect to their effects on oligodendrogenesis in the developing and adult brain [79,80]. BMPs are secreted proteins that signal through heteromeric serine/threonine kinase receptors with BMP2 and BMP4 primarily signaling via the type 1 receptors, BMPR1a and BMPR1b, and the ligand-binding type 2 receptor BMPRII. Both BMPs are highly expressed by qNSCs and TAPs, and their receptors, BMPR1a and BMPRII, are expressed in all stages, but BMPR1b is only expressed by neuroblasts [81]. Active BMP2/4 signaling is only detected in the qNSCs and TAPs in the adult SVZ [82], while noggin, an extracellular inhibitor of BMP signaling, is expressed by the ependymal cells adjacent to the NSCs [81,83].

During development, BMP signaling is a potent inhibitor of oligodendrogenesis, and inhibition of the pathway with noggin rescues this inhibition in vivo and in vitro [84,85,86]. Similarly, inhibition of BMP signaling in qNSCs and TAPs with noggin or in neuroblasts with chordin increases oligodendrogenesis from adult SVZ-derived NSCs in vitro [31,87]. Conversely, an increase in BMP signaling cell-autonomously inhibits proliferation and production of neuroblasts, indicating a functional role for BMP signaling in maintaining quiescence of NSCs and preventing neurogenesis [81]. Consistently, inhibition of BMP signaling in adult qNSCs in vivo, either genetically using Glast-CreER; Smad4-floxed mice or pharmacologically with noggin infusion, significantly reduces neurogenesis by decreasing the number of neuroblasts without affecting the number and proliferation of qNSCs and TAPs. This effect is due to an increase in Olig2 expression in the TAPs, resulting in migration of the neuroblasts to the corpus callosum and differentiation into oligodendrocytes, thereby increasing oligodendrogenesis at the expense of neurogenesis in the healthy adult brain [82].

These effects of BMP signaling on oligodendrogenesis are recapitulated upon demyelination, where a reduction in BMP signaling is accompanied by decreased neurogenesis in the olfactory bulb with a corresponding increase in the proliferation of Olig2+ TAPs, leading to an increase in oligodendrocytes in the lesion [56]. Demyelination increases BMP4 signaling in qNSCs and TAPs but not in neuroblasts. It also increases the number of qNSCs in the adult SVZ, and inhibition of BMP signaling with noggin infusion significantly decreases their number with a concomitant increase in TAPs without affecting the number of neuroblasts [87].

The TGFβ subfamily consists of three isoforms, TGFβ1, TGFβ2, and TGFβ3, which bind to serine-threonine kinase receptors. TGFβ1 signals by binding to the receptor TGFβR2 leading to recruitment and phosphorylation of the TGFβR1 receptor, activating its kinase. Activated TGFβR1 phosphorylates Smad2/3, which form a complex with Smad4 and translocate to the nucleus for regulation of transcription by interacting with smad-binding elements in the promoter regions [88]. TGFβ1 is predominantly expressed in the choroid plexus and meninges in the healthy adult brain, but following an injury, it is expressed by astrocytes, neurons, and microglia. The ligand-binding receptor TGFβR2 is mostly expressed in TAPs along with a smaller proportion of qNSCs, but not in OPCs in the adult brain [89]. Unlike BMP signaling, genetic loss of TGFβ signaling in qNSCs has no effect on the number of TAPs or neuroblasts [82]. However, intraventricular infusion of TGFβ1 decreases the proliferation of adult NSCs [89].

The TGFβ family is of great interest since it influences proliferation, migration, and differentiation of NSCs in addition to modulating the immune system [90]. Indeed, TGFβ ligands, including TGFβ1 as well as the receptors TGFβR1 and TGFβR2, are highly expressed by reactive astrocytes and microglia in chronic MS lesions [91]. TGFβ1 is also upregulated in the aging brain where remyelination is limited [92,93]. In rodents, demyelinating lesions show higher TGFβ1 expression [94], and transgenic overexpression of TGFβ1 in the brain results in earlier onset and more severe disease in the EAE model, suggesting a negative role in remyelination [95]. On the contrary, genetic knockout of TGFβ1 in the developing embryo results in focal demyelination in the brain, suggesting a positive effect on developmental myelination [96]. Thus, the actions of TGFβ1 signaling are dependent on the timing, concentration, and cell type, suggesting the presence of specific modulators of the pathway in different cells to enable a differential response. Indeed, NSCs in the healthy SVZ respond to TGFβ1 by inhibiting proliferation at lower levels and inducing apoptosis at higher levels [89]. However, the effects of TGFβ1 on different subsets of qNSCs for oligodendrogenesis and the molecular mechanisms involved remain unresolved.

#### 2.2.3. Epidermal Growth Factor (EGF)

The expression of the EGF receptor (EGFR) coupled with a high proliferative capacity enables the TAPs to grow as neurospheres in the presence of EGF, in vitro [19,20]. Enhancing EGF signaling by intraventricular infusion of EGF expands the qNSCs, increasing the generation of Olig2+Nestin+ TAPs, which migrate and differentiate into oligodendrocytes in the healthy corpus callosum, septum, striatum, and cortex of the adult mouse brain [97]. However, enhancing EGF signaling specifically in TAPs, by overexpressing EGFR in these cells in vivo, increases their proliferation but reduces the activation of qNSCs, suggesting a role for EGF in regulating the relative numbers of qNSCs and TAPs [36]. These EGF-expanded TAPs also promote remyelination following focal demyelination in the adult brain [97,98]. Indeed, inhibition of EGF signaling in vivo, using a hypomorphic EGFR mutant mouse, decreases the proliferation of TAPs, leading to reduced oligodendrogenesis following focal demyelination in the adult brain [99]. Thus, EGF signaling mainly functions to promote proliferation of TAPs and oligodendrogenesis. Whether this pathway has similar effects on TAPs derived from different subsets of qNSCs remains unknown.

#### 2.2.4. Wnt Signaling

The Wingless/Integrase1 (Wnt) family of secreted glycoproteins consists of 19 Wnt proteins that signal via β-catenin dependent (canonical) or independent (noncanonical) pathways. Canonical Wnt signaling involves stabilization and translocation of β-catenin to the nucleus, where it binds TCF/LEF transcription factors to activate downstream effectors of the pathway, including Axin2, a sensitive readout of the pathway [100]. Wnt/β-catenin signaling is active in qNSCs and TAPs but not in the neuroblasts of adult SVZ [38,101]. It has different outcomes in the dorsal SVZ, which is more oligodendrogenic compared to the more neurogenic lateral SVZ. In the lateral SVZ, activation of β-catenin increases neurogenesis in the olfactory bulb by enhancing proliferation of TAPs [101,102]. In the dorsal SVZ, β-catenin is expressed mostly in qNSCs, with relatively fewer TAPs and neuroblasts in the adult brain [103]. Activation of the pathway selectively stimulates proliferation of the oligodendrogenic lineage by reducing the cell cycle length, ultimately generating a higher number of oligodendrocytes from the dorsal SVZ [59]. Thus, canonical Wnt signaling enhances oligodendrogenesis from adult NSCs in the dorsal SVZ but increases neurogenesis in the lateral SVZ.

In contrast to the canonical pathway, non-canonical Wnt signaling acting via Cdc42 promotes quiescence in qNSCs by inhibiting proliferation [104]. However, demyelination activates the canonical pathway and down-regulates the non-canonical pathway, which activates qNSCs, induces proliferation of TAPs, and increases oligodendrogenesis [104]. Canonical Wnt pathway is also active in MS lesions in the human brain; however, dysregulated pathway activity ultimately leads to failure of remyelination by inhibiting maturation of OPCs into myelinating oligodendrocytes [105]. Overall, canonical Wnt/β-catenin signaling increases proliferation and enhances oligodendrogenesis from select NSC populations in the adult SVZ.

#### 2.2.5. Notch

Notch is a transmembrane protein that binds to receptors Dll1 and Jagged1 in neighboring cells, leading to cleavage and translocation of the notch intracellular domain (NICD) into the nucleus where it activates target genes such as Hes1 and Hes5 [106]. Notch1 and Hes1 are predominantly expressed in qNSCs, whereas Dll1 and Jagged1 are expressed in TAPs and neuroblasts. Increasing notch1/Jagged1 signaling increases neurogenesis by stimulating the proliferation of TAPs [36,107,108]. The qNSCs become dependent on notch signaling upon activation, and thus, ablation of notch signaling reduces neurogenesis without decreasing the number of qNSCs [109,110]. This pathway also interacts with other signaling pathways such as Shh and EGF and it improves the survival of qNSCs by activating Shh and mTOR pathways [111]. However, EGF signaling in TAPs reduces notch activity in qNSCs, inhibiting their proliferation [36]. Thus, notch activity regulates neurogenesis mainly by facilitating communication between the qNSCs and TAPs in the healthy adult brain.

Demyelination induces Jagged1 expression and increases notch activity in Olig2+ TAPs and OPCs [112]. It also upregulates Endothelin1 (ET-1), which activates notch signaling in qNSCs, inducing their quiescence and reducing neurogenesis [113]. Overall, activation of Notch signaling via noncanonical Wnt signaling or direct overexpression of the notch intracellular domain (NICD) inhibits remyelination by promoting NSC quiescence and decreasing oligodendrogenesis [104].

## 3. Subgranular Zone (SGZ)

The SGZ is the layer of cells, adjacent to the granule cell layer, lining the hilus in the dentate gyrus of the hippocampus and is one of the principle areas of neurogenesis along with the SVZ. Since it is located away from the ventricles, it is not in close proximity to ependymal cells. However, similar to the SVZ, the radial glia-like NSCs, with their cell bodies in the subgranular layer and their processes traversing the granule cell layer, give rise to intermediate progenitors or transit-amplifying progenitors (TAPs) and neuroblasts that generate granule neurons [114]. The NSCs express GFAP and Sox2 while the TAPs express Sox2, Olig2, and proliferation markers such as MCM2, and the neuroblasts express PSA-NCAM and doublecortin (Dcx) [115,116,117]. The fate of NSCs in the SGZ can be altered by overexpressing or inhibiting transcription factors. As observed in the SVZ, Pax6 promotes neurogenesis, and overexpression of Pax6 induces maturation of the neuroblasts into postmitotic neurons. In contrast, inhibition of Olig2, the oligodendrogenic transcription factor, represses oligodendrogenesis and switches the fate of neuroblasts to the astrocytic lineage [118]. In addition, overexpressing Ascl1 inhibits neurogenesis and induces oligodendroglial lineage commitment [119].

Amongst the qNSCs in the SGZ are the Shh-responsive Gli1-expressing cells, with about 19% of these cells being GFAP+ qNSCs and 38% being PSA-NCAM+ neuroblasts [66]. Shh is expressed in the granule neurons of the dentate gyrus, adjacent to the hippocampal Gli1 stem cells (Figure 2) [120,121,122,123]. In vivo overexpression of Shh increases the proliferation of NSCs, and pharmacological inhibition of Shh with cyclopamine reduces their proliferation, although the specific population of NSCs affected is unknown [123].

The SGZ NSCs are regulated by some of the same signaling pathways as the SVZ. While Wnt and Shh pathways activate the NSCs, BMP and notch signaling promote their quiescence [124]. Canonical Wnt signaling via β-catenin is active in the SGZ, with the adult hippocampal NSCs responding to Wnt ligands, i.e., Wnt3 secreted by the astrocytes. Wnt signaling specifically increases neuronal differentiation of NSCs without altering the generation of astrocytes and oligodendrocytes. This effect is due to an increase in proliferation of Dcx+ neuroblasts, but the oligodendrocyte progenitors are not affected by Wnt3 [125].

BMPs are also expressed in the SGZ along with their receptors and their inhibitor, noggin [126,127,128,129]. BMP signaling is critical in maintaining hippocampal neurogenesis, with enhanced BMP signaling decreasing neurogenesis and inhibition of BMP signaling with noggin increasing the self-renewal of GFAP+ NSCs in the SGZ [130]. Similar to that observed in the SVZ, infusion of TGFβ1 in the ventricles or overexpression of TGFβ1 in NSCs reduced their proliferation by arresting them in the G0/G1 phase of the cell cycle, resulting in a decrease in the number of Dcx+ neuroblasts [89,131,132]. Consistently, reducing TGFβ1 signaling in nonhuman primates resulted in an increase in proliferation of NSCs and accelerated neurogenesis in the hippocampus [133].

Hippocampal demyelination is common in people with MS and correlates with memory impairments and cognitive decline [134,135,136]. In mice, demyelination using a cuprizone diet reduced the proliferation of TAPs in the adult SGZ [137,138,139], but the neuroblasts proliferated and differentiated into mature oligodendrocytes [140]. Further studies are needed to identify mechanisms for enhancing remyelination by NSCs in the SGZ.

## 4. Tanycytes Lining the Third Ventricle

The NSCs in the adult hypothalamus play an important role in regulation of food intake and energy balance [141,142]. The hypothalamic third ventricle is lined by a single layer of cells, consisting of multiciliated ependymal cells along with tanycytes, which have a single cilium and a radial glia-like morphology with long processes extending into the parenchyma [143,144]. Based on their position in the third ventricle, the tanycytes are classified as α-tanycytes in the lateral wall and β-tanycytes in the floor, lining the median eminence. Depending on the projection of their processes, α-tanycytes are further subdivided into α1-tanycytes extending from ventromedial to the dorsomedial nucleus and α2-tanycytes adjacent to the arcuate nucleus [145]. The tanycytes express NSC markers such as GFAP, GLAST, vimentin, and nestin; proliferate in vivo; and grow as FGF and EGF-responsive neurospheres, which can differentiate into neurons, astrocytes, and oligodendrocytes in vitro [143,146,147,148,149]. The adult human brain also contains NSCs expressing GFAP and nestin in the hypothalamic third-ventricular wall [40]. Further, in vivo lineage tracing experiments in the adult mouse brain show that tanycytes can generate neurons and astrocytes in the adult hypothalamus, indicating their multipotent capacity [143,145,146]. Thus, these cells can potentially behave as NSCs of the third ventricle.

Tanycytes are regulated by signaling pathways such as Shh and BMPs. Sonic hedgehog (Shh) is known to be important for the development of hypothalamus, and treatment of embryonic hypothalamic neurospheres with Shh in vitro increases their proliferation [150,151]. In addition, permanently labeling the Sonic hedgehog (Shh)-expressing cells in the Shh-CreER mouse embryo (E7.5-E11.5) showed that these cells generate tanycytes in the adult brain [152]. While Shh is expressed in the tanycytes lining the ventral third ventricle, Ptc, Smo, and Gli1 are expressed in the tanycytes lining the median eminence in addition to the ventral part of third ventricle (Figure 2) [122]. In contrast, BMPs and noggin are expressed in the paraventricular hypothalamic nuclei surrounding the third ventricle [127,128,129]. However, the role of Shh and BMPs in oligodendrocyte generation in the healthy hypothalamus and in remyelination is unknown.

Although tanycytes do not generate oligodendrocytes in the healthy adult brain, indirect evidence suggests that these cells may be activated in response to demyelination in the hypothalamus and the neighboring optic nerve. Demyelination of the optic nerve close to the third ventricle increased the proliferation and generation of oligodendrocytes [153], consistent with their role in myelination of the optic nerve during development [154]. Moreover, when NSCs harvested from the median eminence containing the β-tanycytes were transplanted into the shiverer mouse brain lacking myelin, they preferentially differentiated into oligodendrocytes [155]. Thus tanycytes residing in the third ventricle are a potential source of NSCs for remyelination.

## 5. Cerebellum

The adult cerebellum contains NSCs that form neurospheres and differentiate into neurons, oligodendrocytes, and astrocytes in vitro, but their precise in vivo location is unknown [156]. Radial glia are precursors of NSCs in the brain [157], and the cerebellum contains a discrete population of these cells called Bergmann glia, with their cell bodies located in the Purkinje cell layer and processes (Bergmann fibers) extending radially to the pial surface of the adult cerebellum [158]. Based on co-expression of NSC markers such as Sox1/2/9, GFAP, GLAST, and vimentin, several studies have suggested that Bergmann glia are the NSCs in the adult mouse and human cerebellum [159,160,161,162]. Moreover, these cells proliferate in response to exercise [161] and death of granule cells [163]. They express Gli1, and their cell bodies are adjacent to the Purkinje neurons, which secrete Shh (Figure 3) [121,164,165]. In fact, Shh is essential for generation of Bergmann glia from precursor cells in the developing cerebellum [166]. In addition, notch signaling is essential for their position and morphology in the adult cerebellum [167]. BMP2 and BMP4 are also expressed in Purkinje neurons and Bergmann glia, but their functional role is unclear [128,129]. Thus, Bergmann glia are likely to be the putative NSCs of the adult cerebellum, but their role in oligodendrogenesis in the healthy cerebellum and in remyelination needs further examination.

The nature and source of myelin regeneration in the adult cerebellum are still not clear, despite the description of cerebellar signs in people with MS as early as 1877 by Charcot. Cerebellar dysfunction, early in the MS disease course, is predictive of disability and disease progression, and most people with MS develop cerebellar signs after they enter the progressive stages [168,169,170,171,172]. Overall, 11–33% of people with MS are predominantly affected by cerebellar signs and symptoms, which include tremors, ataxia, and slurred speech, underscoring the importance of enhancing remyelination in the cerebellum [173,174]. Moreover, the cellular source and mechanisms of remyelination are likely to be different from those in the cerebrum, as indicated by the lack of effects by the FDA-approved MS drug fingolimod in the cerebellum [175].

## 6. Conclusions and Future Perspectives

Oligodendrogenesis is essential in the healthy brain for replacing worn-out oligodendrocytes and for promoting myelin plasticity. It is also necessary for myelin regeneration following demyelinating injury in the adult mammalian brain. There are distinct pools of NSCs that generate oligodendrocytes with distinct molecular mechanisms of oligodendrogenesis in the healthy vs. demyelinated brain. Future studies using genetic manipulation of signaling pathways in specific subsets of NSCs are essential to delineate the molecular mechanisms regulating oligodendrogenesis in the healthy vs. demyelinated brain.

There are considerable differences between remyelination achieved by new oligodendrocytes derived from OPCs and NSCs. Most notably, axons remyelinated by NSCs have a normal G-ratio in contrast to thinly myelinated axons, which are a hallmark of remyelination by OPCs [42,57]. In addition, there are potential differences between remyelination by aged local OPCs vs. newly formed OPCs by NSCs, consistent with limited remyelination by OPCs observed in the aging brain [176,177]. While aged OPCs have limited capacity for remyelination, NSCs retain their potential for proliferation and differentiation in the aging brain, although their numbers decline with age, thus underscoring the need to enhance remyelination by NSCs [178]. More importantly, inhibition of NSC-mediated remyelination by ablating them results in axonal loss highlighting their significant functional contribution to remyelination [179]. Taken together these studies suggest that remyelination by NSCs may provide chronic axonal protection, and enhancing remyelination by NSCs may be beneficial to preventing axonal degeneration thus providing a strategy for neuroprotective therapy in MS and other neurological diseases.

Finally, the potential for enhancing oligodendrogenesis from NSCs residing in multiple areas of the adult brain is becoming increasingly clear. In order to harness this potential, we have to not only decipher how NSCs in each niche are regulated by the different signaling pathways but also take into consideration the heterogeneity of NSCs within each niche.

## 7. Patents

A patent on the method of targeting GLI1 as a strategy to promote remyelination has been awarded, with J. Samanta listed as a co-inventor.

## Figures and Tables

**Figure 1 cells-11-02101-f001:**
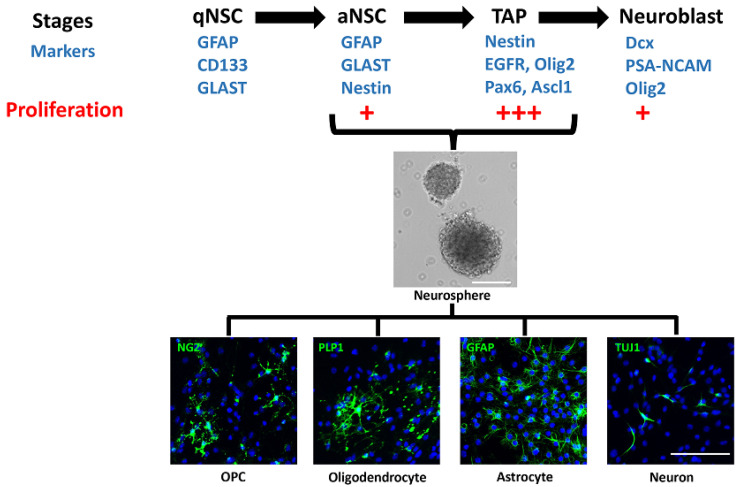
Stages of neural stem cells and their markers. The different stages of neural stem cells (NSCs) are identified by expression of markers and level of proliferation. Only activated NSCs (aNSCs) and transit-amplifying progenitors (TAPs) grow as neurospheres and differentiate into the oligodendroglial, astroglial, and neuronal lineages in vitro. OPC, oligodendrocyte progenitor cell.

**Figure 2 cells-11-02101-f002:**
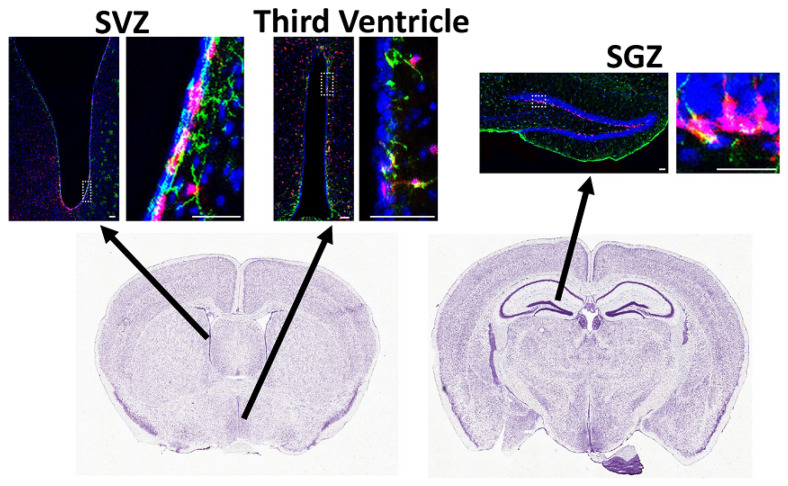
Neural stem cells in the adult subventricular zone (SVZ), hippocampal subgranular zone (SGZ), and the lining of the third ventricle. Gli1 NSCs were fate-mapped by tamoxifen injection in the adult (P60-80) Gli1CreER;Ai9 mice and analyzed by immunofluorescence for GFAP (green) to detect NSCs, TdT (magenta) to detect Gli1 fate-mapped cells, and Hoechst (blue) to detect cell nuclei, 2 weeks after tamoxifen administration. The right panels are higher magnification views of the dotted boxes in the left panels. The arrows indicate the area in the coronal brain sections from Allen brain atlas. Scale bar = 50 μm.

**Figure 3 cells-11-02101-f003:**
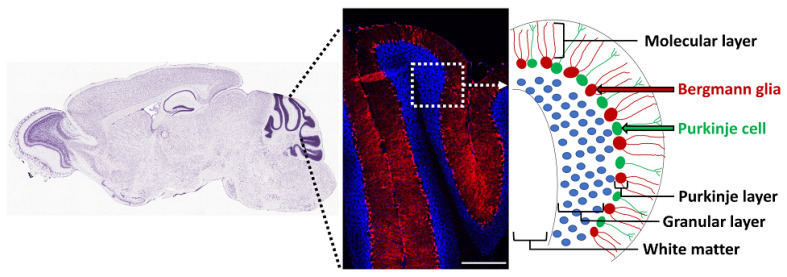
Bergmann glia in the adult cerebellum. Gli1-expressing Bergmann glia were fate-mapped by tamoxifen injection in the adult (P60-80) Gli1CreER;Ai9 mice and analyzed by immunofluorescence to detect TdT-positive (red) Gli1 fate-mapped cells and Hoechst (blue)-positive cell nuclei, 2 weeks after tamoxifen administration.

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
