# Peer review of "Endogenous Neural Stem Cell Mediated Oligodendrogenesis in the Adult Mammalian Brain"

_cells, 2022, doi:10.3390/cells11132101_

Round 1
Reviewer 1 Report
Cells, Review
Endogenous neural stem cell mediated oligodendrogenesis in the adult mammalian brain.
Daniel Z. Radecki and Jayshree Samanta
Neural stem cells in the adult mammalian brain constitute an endogenous source of multipotent cells. They continuously generate new neurons in two restricted niches: the subventricular zone (SVZ) of the forebrain lateral ventricles, and the subgranular zone (SGZ) of the dentate gyrus in the hippocampus. Radial glia-like stem cells give rise to amplifying progenitor cells, which mostly become neurons. In addition, multipotent NSCs can generate glia (in vivo and in vitro). NG2 (OPCs) progenitor cells also reside in the niches as a potentially multipotent subpopulation that makes no apparent contribution to adult neurogenesis. Recently, the hypothalamus has emerged as a third region of postnatal neurogenesis and gliogenesis. Populations of specialized radial glial tanycytes are thought to regulate the hypothalamic in- and output of circulating hormones and nutrients. However, only a small number of newborn cells become neurons. In this review, Radecki and Jayshree summarize the role of NSC in theses niches, incl. the cerebellum, as source for oligodendrogenesis. It is an interesting and elaborate manuscript, describing the different niches, signaling factors, and the function of new born oligodendrocytes associated with disease models.
Please see specific comments and questions below as they might strengthen the review.
Abstract and Introduction
Li 8, 9 repeat this sentences similarly throughout, discussion and conclusion
Li 22,23 reverse the sentence, such as “In the CNS, oligodendrocytes’ main function is the myelination of axons”; the authors could also mention its role in de-, or remyelination
Li 37, ‘by adult oligodendrocytes’ , instead of pre-existing (?)
The last paragraph (li 50-63), please distinguish in vitro vs in vivo
2. SVZ, li 65 ‘.. consist of V-SVZ and SEZ..’ instead of ‘known’; what does ‘activation, reactivation’ means throughout this paragraph? Li 69 ‘NSCs give rise to ..’ ; li 86,87 move up to 80, li 104, migration of neuroblast (RMS) towards the olfactory bulb should be mentioned ;
2.1 please check the order, words of the first sentences (li 116-121), and maybe provide addition headlines a) for retrovirus detection of lineage progression, b) development (from line 155-178 – these 2 paragraphs could also be shortened , merged with line 111-114), c) Gli1 expressing cells
3. SGZ, 367, ‘The SGZ is the inner layer of the granule cell layer lining the hilus ..’; li 368 ‘.’ Behind ependymal cells.’ Li 371‘… similar to the SVZ, radial glia-like cells give rise to transient-amplifying …’ li 373, please see your citation 112 for marker expression; take out the last sentence, or describe better what you mean; for glia expression (such as NG2, Rip) please refer to Klempin et al 2012 or Palmer et al 1999
4. li 407 please start with the Stem Cell population rather than ‘neurogenesis’ since this is still in debate (the majority of studies shows only a few newborn neurons)
5. here you start with the disease, consider revising the paragraphs; see minor comments below, take out li 455, and start with ‘BG cell bodies..’
6. li 477 start with the importance of oligodendrogenesis in the adult brain; their important function in disease
Figure 1, IHC picture could be bigger
Figure 2, rather provide a coronal brain section as overview image, in line with IHC
Figure 3, the overall brain slide could be a little bigger
You could also provide a summary Table showing the signaling factors (2.2)
Minor comments, please check abbreviations throughout, e.g. li 75 CSF, and NSC, BG 453 - 69, shorten sentences.
Reviewer 2 Report
This is a timely review on oligodendrogenesis in the adult mammalian brain. The authors thoroughly review recent literature of the field and describe novel findings that may lead to new areas of interest. First, they describe the process of oligodendrogenesis from NPCs in the subventricular zone and explain relevant signaling pathways that regulate oligodendrogenesis (such as Shh, TGF beta, etc); then the authors describe oligodendrogenesis in the subgranular zone of the hippocampus and tanycytes of the third ventricle; finally a description of oligodendrogeneisis in the adult cerebellum is briefly explained; considering the absence of detailed research on olgodendrogenesis in this area of the brain, the authors provide important references of what is currently known of this subject.
In general, the review is well written and I have no doubts that it will provide important information to colleagues interested in this field. I have three minor suggestions: 1) L46, please provide more reference regarding the role of parenchymal OPCs in the adult – only two are listed [7 and 8], this may lead to the interested readers to important findings in this field, 2) Figure 2. The drawing supposes to illustrate Bergmann glial cells, instead the cells drawn appear like Purkinje neurons – large soma and dendrites – redraw that scheme since it is misleading, 3) my version of the manuscript shows figures in low resolution, in particular I find fluorescence images are blurred, please work with the editorial house to improve the presentation of final images. Write Purkinje with uppecase.
